# What Happened to the Phycobilisome?

**DOI:** 10.3390/biom9110748

**Published:** 2019-11-19

**Authors:** Beverley R. Green

**Affiliations:** Botany Department, University of British Columbia, Vancouver, BC V6N 3T7, Canada; brgreen@mail.ubc.ca

**Keywords:** chloroplast, primary endosymbiosis, historical contingency, phycobilisome, LHC, prochlorophyte, chlorophyll, phycobiliprotein

## Abstract

The phycobilisome (PBS) is the major light-harvesting complex of photosynthesis in cyanobacteria, red algae, and glaucophyte algae. In spite of the fact that it is very well structured to absorb light and transfer it efficiently to photosynthetic reaction centers, it has been completely lost in the green algae and plants. It is difficult to see how selection alone could account for such a major loss. An alternative scenario takes into account the role of chance, enabled by (contingent on) the evolution of an alternative antenna system early in the diversification of the three lineages from the first photosynthetic eukaryote.

## 1. Introduction

Gene loss is a major factor in evolutionary change [1,2,3]. While new genes are continually being acquired by duplication, recombination, horizontal transfer, endosymbiotic transfer, and de novo origin, other genes are being lost by the gradual accumulation of deleterious mutations (pseudogenization), insertions, deletions, and chromosomal reorganization. These losses are a major contributor to the pool of genetic variation. Certain losses can provide a selective advantage in adaptation to novel environmental challenges, e.g., drought tolerance in wild Arabidopsis [4] and thermophily in basal red algae [5]. Losses of genes and gene families follow episodes of whole-genome duplication and are correlated with speciation [3].

We tend to think about adaptation in terms of genetic changes being acted upon by positive or negative selection over time, and consequently spreading through the resulting population, although many genetic changes appear to be neutral or nearly neutral. How could selection alone account for the loss of an entire macromolecular complex like the phycobilisome? The phycobilisome (PBS) is a highly efficient light-harvesting complex essential for photosynthesis in cyanobacteria, rhodophyte algae, and glaucophyte algae [6,7,8,9]. but it has been completely lost in the chlorophyte lineage (Figure 1). Plants and green algae depend only on membrane-intrinsic Chl *a/b* proteins, members of the light-harvesting (LHC) superfamily, while the red algae have both PBS and Chl *a*-binding members of the LHC family [8,10].

What made it possible for one of the descendants of the first photosynthetic eukaryote to survive the loss of the PBS? Could selection alone account for this, or was it a loss that only happened to succeed because of pre-existing factors in the organism’s history, such as certain mutations or particular environmental conditions at that time [11]? This is the concept of historical contingency, put forth by Steven J. Gould as a major factor in evolution [12]. Expressed in another way, a certain outcome is contingent on (depends on) the details of evolutionary history, so that there could have been a different outcome under different starting conditions [11,13]. The question: “Can the Tape of Life be replayed?” is being actively investigated at the molecular level using ancestral gene reconstructions combined with comparative structure/function analysis and long-term in vitro evolution experiments [13,14,15].

## 2. Phycobilisome Basics

The phycobilisome is generally a very large and elaborate macromolecular structure made up of pigment-binding proteins that absorb visible light, and funnel that energy into photosynthetic reaction centers to drive the central processes of photosynthesis [6,7,16,17]. It is the major light-harvesting antenna in cyanobacteria and two of the three phyla of photosynthetic eukaryotes with primary plastids. Unlike photosynthetic reaction centers and all other light-harvesting antennas, the phycobilisome chromophores are linear tetrapyrroles (phycobilins) rather than chlorophylls (Chls). Different phycobilins absorb light at different wavelengths due to small structural differences. Each type of phycobilin is bound via thioether linkage(s) to specific members of the large phycobiliprotein family which evolved via successive rounds of gene duplication [18]. The way in which the different combinations of pigments and proteins are organized into a complex macromolecular structure is what makes the phycobilisome so effective in absorbing and transferring light.

The protein building block of a phycobilisome is assembled from pairs of α and β subunits, each with 1–3 attached phycobilins, which associate to form disc-like trimers, then stack face-to-face to form hexamers. Hexamers are assembled into rods of different types with the aid of linker molecules. The “model” cyanobacterial phycobilisome shown in Figure 2A has a core of three short rods of allophycocyanin (APC). Attached to this core are a number of rods, each made up of one or more phycocyanin (PC) hexamers. In some species, these rods are extended by the addition of phycoerythrin (PE) hexamers. While hexamers can form spontaneously, their assembly into rods is dependent on a number of linker proteins; different ones for PC, APC, and PE rods, and yet others that join one type of rod to another. The PBS is attached to the thylakoid membrane by the trans-membrane helix of another linker (Lcm), a special APC unit in the core.

The key to the operational efficiency of this common structure is that its elements absorb and emit light energy at different wavelengths and transfer energy down a cascade of decreasing energy, shown schematically in Figure 2B. PE (the red colored pigment of red algae) absorbs in the blue-green region of the spectrum (the highest energy light), transfers it to PC (which absorbs in the orange region), from there to APC, then to the core linker Lcm, and finally to the reaction center. The final destination is primarily photosystem II (PSII), but some energy can be transferred to photosystem I (PSI) by a number of (still debated) mechanisms under physiological control [16,17]. This may involve formation of mega-complexes where both PSII and PSI are under the umbrella of the (relatively large) PBS [19]. However, the PSI reaction center complex (PsaA/PsaB) has a much larger number of core antenna Chls than PSII, so from a functional point of view, the PBS is largely a PSII antenna. The PBS of red algae have the same general structure as those of cyanobacteria and glaucophytes, although they tend to be larger, with peripheral rods made up mainly of PE [8].

Although a number of high resolution x-ray crystal structures of various hexamers have been published since the 1990s, and the overall organization of red and cyanobacterial PBS has been determined with a combination of biochemical and electron microscopy techniques [7,20], attempts to obtain a good three-dimensional structure of a whole PBS were frustrated by the complicated nature and interactions of the linkers. The first high resolution three-dimensional structure to show the linkers and their interactions was that of the red alga *Griffithsia pacifica*, only recently determined by cryo-electron microscopy (Figure 3) [21]. This breakthrough was all the more amazing since its PBS is enormous, with a total molecular weight of 16.8 megadaltons, including 862 proteins and 2048 chromophores. There are 5 APC rods in the core and 14 peripheral rods made up mainly of PE, but some with one PC hexamer bridging to the core, and some individual hexamer units. The model revealed the complicated ways in which the 72 linkers make up the skeleton of the PBS by binding in the hollows between hexamers with extensions to neighboring linkers and hexamers. How this structure is assembled and disassembled correctly is still a puzzle.

It was this structure that motivated me to ask the question “What happened to the phycobilisome?” and write this review. What factors drove the loss of such a complex and highly effective light-harvesting structure as the PBS? Even though the PBS of the cyanobacterial endosymbiont may have been much simpler than those of the extant red and glaucophyte algae, what sort of selective advantage (if any) could the loss of such a structure possibly have provided?

## 3. What a Few Deviant Cyanobacteria Might Tell Us about PBS Loss

Several unrelated genera of cyanobacteria appear to have lost the PBS [22,23]. *Prochloron* and *Prochlorothrix* have only PC hexamers and no linkers to connect them into rods. Low light-adapted strains of *Prochlorococcus* have PE rod linkers and make short detached rods, while the high light-adapted strain has lost everything except a very degenerate PE-β gene. It is unclear how much these remnants can contribute to the energy economy of the cell. Most members of a fourth genus, *Acaryochloris*, make rods of four PC hexamers that may be able to transfer a small amount of energy to PSII [24].

In every case, the function of the PBS has been replaced by Chl-binding membrane-intrinsic proteins (Pcbs) related to the IsiA protein, which was originally discovered as a Chl *a*- protein induced by Fe limitation in *Synechocystis* 6803. Under these conditions, IsiA forms a ring of 18 units around PSI, which results in absorption of more than enough light to compensate for the decrease in PSI reaction centers provoked by low Fe [22]. IsiA proteins can also be induced by high light or oxidative stress. The various members of the Pcb-IsiA family evolved via successive rounds of duplication and divergence from the *psbC* gene, which encodes the apoprotein of CP43, one of the core antennas of PSII [25,26,27]. IsiA homologs have been found in 125 out of 390 sequenced cyanobacterial strains [28], all of which have PBS and are believed to synthesize Chl *a* only. A phylogenetic analysis of a smaller number of genome sequences revealed several previously unrecognized branches of the IsiA-Pcb family [29], but so far there is no functional information about their biological role.

In *Prochloron* and *Prochlorothrix*, the Pcb proteins bind Chls *a* and *b*, and in *Prochlorococcus* they bind divinyl-Chls *a* and *b*. *Acaryochloris* Pcb proteins bind Chl *d*, since these cells use Chl *d* for all their photosythetic proteins, which enables them to live in low light environments enriched in infrared light [24]. However, the Chl *a/b* antenna proteins in the first three genera (often collectively referred to as prochlorophytes) are completely unrelated to the Chl *a/b* proteins of the green algae and plants [25,26,27]. Pcb proteins form an 18-membered ring around PSI constitutively in *Prochlorococcus* and under Fe starvation in *Acaryochloris*; eight Pcb proteins flank each PSII dimer in *Prochloron* and *Acaryochloris* [22].

These four unrelated groups of cyanobacteria live in very different habitats. *Prochloron* is a symbiont of didemnid ascidians in coral reefs, the few known species of *Prochlorothrix* live in freshwater lakes, and *Acaryochloris* lives underneath ascidians harboring *Prochloron* or under other algae that absorb most of the incoming radiation and leave it enriched in near-infrared light. *Prochlorococcus* is the only one that is widespread: it is a major component of the phytoplankton in many parts of the ocean [30,31]. It is hard to discern any common selective pressure that would have favored the loss of the PBS in these unrelated and environmentally disparate cyanobacteria. What it does suggest is that in each case, the gradual loss of the PBS was enabled by the availability of an alternative light-harvesting antenna based on Chl *a* and a Pcb-IsiA homolog. In other words, the loss was contingent on the presence of a pre-existing gene or genes that could diversify into an alternative antenna family.

## 4. Light-Harvesting in the Three Primary Lineages of Photosynthetic Eukaryotes

There is general agreement that a single primary endosymbiosis involving a cyanobacterial symbiont and a heterotrophic host gave rise to the first photosynthetic eukaryote with a plastid [32]. This new type of eukaryote subsequently diversified over more than a billion years to produce the three extant lineages: the Chlorophyta (green lineage), the Rhodophyta (red lineage), and the Glaucophyta (Figure 1). Much can happen in a billion years, and the first photosynthetic eukaryote would surely have given rise to a number of lineages besides these three. Some would have evolved interesting innovations that cannot even be imagined, but unfortunately, they are no longer available to be studied. Too much time has elapsed to determine which (if any) of the currently existing cyanobacteria are the closest relatives of the ancestral plastid, although this continues to be an active topic of discussion [33].

The first photosynthetic eukaryote would have retained its cyanobacterial phycobilisome and would have had the machinery to make Chl *a*, and possibly other Chls [34]. If this cyanobacterial ancestor had *IsiA* genes, they have been lost in all of its descendants including the Glaucophyta. There is also no trace of the cyanobacterial orange carotenoid protein (OCP) in any of the three lineages. The OCP binds to the PBS core under high light and releases excess excitation energy as heat, thus preventing photodamage to the PBS, and indirectly to PSII [35].

The first photosythetic eukaryote did retain something very important from the cyanobacterial ancestor: the genes for small proteins with one transmembrane helix (called Hlips or OHPs), potentially binding both Chl *a* and carotenoids, and originally discovered as high light-induced proteins. In cyanobacteria they are important for photoprotection via their role in assembly and repair of PSII but may have other functions (reviewed in [36]). Genes encoding Hlips are found in all photosynthetic eukaryotes, with copies on both the plastid and nuclear genomes (red and glaucophyte algae) or just in the nuclear genome (chlorophytes) [37]. Their sequences are clearly related to the first and third helices of the membrane-intrinsic LHC proteins of red and green lineages. This suggests that a primitive three-helix Chl-protein which bound carotenoids and Chl *a* (the proto-LHC) evolved by tandem duplication of a nuclear Hlip gene, with inclusion of intervening sequence that could form a middle transmembrane helix and put the first and third helices in position to bind Chls betweeen them like the LHCs [38,39]. Starting from this modest beginning, the precursor gave rise to all the three-helix members of the LHC family, which duplicated and diversified independently after separation of the red and green lineages [40,41].

With phycobilisomes as well as LHCs, the red algae are the only group that efficiently harvests light across the complete visible range (400–700 nm). Their LHC proteins (called Lhcrs) bind only Chl *a* and the carotenoid zeaxanthin, and are associated only with PSI [8]. The five red algae so far investigated have 3–12 genes encoding a small family of closely related Lhcr proteins [41,42]. Two recent high-resolution structures of PSI in *Cyanidioschyzon merolae* showed that the Lhcrs bind to similar positions on the PSI core complex, as do the Lhca proteins of PSI in the green lineage [43,44].

In contrast to the red algae, the green lineage completely lost the PBS but extensively diversified the LHC family [23,39,41]. Certain Chl *a/b* LHCs are associated specifically with PSII (CP26, CP29, CP24, encoded by Lhcb4–6) or PSI (LHCI, encoded by Lhca1-4 in plants; Lhca1-6 in green algae). A large fraction of the light-harvesting is done by trimers of Lhcb1-3 (LHCII) that can move in the membrane plane to transfer energy to either photosystem. The Chl *a/b* LHC proteins bind carotenoids that moderately extend the range of wavelengths absorbed but do not fill the green gap left by the loss of the PBS. However, their other major role is to convert excess excitation energy into heat, and thus protect PSII from photodamage. There are a number of other members of the family, some of which specialize in photoprotection rather than light-harvesting, reviewed in [37,45,46]. The dual roles of photoprotection and light-harvesting were particularly important in the transition to land because the higher oxygen levels and desiccation stress increase the risk of photodamage from the formation of reactive oxygen species.

No three-helix LHCs are found in the Glaucophyta, although they have Hlip genes on both plastid and nuclear genomes [10,37]. The PBS is their only light-harvesting antenna. There is no current agreement about the branching order of the three lineages (the reason for the somewhat ambiguous dotted line in Figure 1), but if the glaucophyte lineage branched off first, the LHC family could have originated in the common ancestor of the red and green lineages. Alternatively, if the common ancestor of all three lineages had already evolved a primitive LHC, the glaucophytes could just as well have lost it early in their evolution. The fact that there are no photosynthetic eukaryotes without either a PBS or an LHC family shows that loss of the PBS was contingent on the pre-existence of at least a small LHC family very early in the evolution of the green lineage.

## 5. Advantages and Disadvantages of Losing the PBS

What other factors besides chance could have been involved in the loss of the PBS and the survival of photosythetic eukaryotes without it? Nitrogen is a limiting resource in most environments. Stadnichuck and Tropin [47] have pointed out that the LHC proteins are more economical than the phycobiliproteins in terms of the number of amino acids required to bind a chromophore. Using their figures, this comes out to 4–6 chromophores per 100 amino acids for the average LHC protein in either lineage, compared to only one chromophore per 100 amino acids for the average phycobiliprotein (not counting the non-pigmented linkers). So, in terms of basic nitrogen demand, LHCs are cheaper than PBSs.

The PBS appears to be a very complex structure to synthesize (i.e., more than a little “assembly required”), especially considering the very complicated linker network that holds it all together [21]. However, many studies have shown that the PBS disassembles very readily under nitrogen deprivation in both cyanobacteria and red algae, and can be quickly reassembled once a suitable nitrogen source is available [48,49]. The PBS can thus be considered a replenishable nitrogen store, not just a light-harvesting complex. This flexibility could provide a significant advantage in periodically nitrogen-limited environments, since the membrane-intrinsic LHCs are much slower to turn over.

The red and green lineages differ substantially in the mechanisms for balancing the energy arriving at the two photosystems. In the green lineage, light preferentially absorbed by PSII causes phosphorylation of a fraction of LHCII which detaches from PSII and migrates to PSI, resulting in a larger fraction of energy being transferred to PSI and relieving the overexcitation of PSII, a phenomenon referred to as a state transition (reviewed in [50]). However, in red algae there is no phosphorylation of the Lhcrs, which are constitutively attached to PSI, although there has long been evidence for direct energy transfer from PBS to PSI [8,50,51]. PBS are not phosphorylated, but they can diffuse in the membrane plane in mesophilic red algae, although not in the thermophilic *C. merolae* [52]. The story has been further complicated by recent evidence for a megacomplex of PBS-PSII-PSI in cyanobacteria [19], as portrayed in Figure 2A, and evidence that such a megacomplex may exist (if only transiently) in at least one red alga [51].

There are significant differences in how the different types of plastid deal with excess excitation energy. In the green lineage, high light exposure and other stresses that slow down the operation of the electron transport chain trigger the induction of the xanthophyll cycle, where violoxanthin is converted to zeaxanthin, a carotenoid very effective at thermal dissipation, and converted back again under low light or darkness. At the same time, special members of the LHC family, such as PsbS and LHCSR, that are particularly well structured for energy dissipation are induced. A variety of mechanisms are involved, collectively referred to as non-photochemical quenching (NPQ); details can be found in the cited references [50,53,54]. Overall, the LHC superfamily is extremely effective at balancing efficient light-harvesting with photoprotection.

At first glance, the red algae would seem to be at a disadvantage. The PBS is the only light-harvesting complex without carotenoids, and red algae do not have the orange carotenoid protein that protects the PBS (and PSII) in cyanobacteria [35]. They do not have a xanthophyll cycle or specialized members of the LHC family, although they do demonstrate increased energy dissipation under high light, which has been variously ascribed to PBS motility, reaction center quenching, or relaxation of the proton motive force [51,52,55,56]. However, a number of studies have shown that the red algae are actually very resistant to the effects of high light exposure [8].The large number of *Ohp* genes in intertidal species such as *Porphyra* [42] suggests these proteins could be involved in some way, but that has not been demonstrated experimentally.

Considering all the factors that have to do with acclimation and adaptation, it is clear that both green and red lineages have evolved a variety of mechanisms to cope successfully with their modern environments. However, we have little information about the environment(s) in which the two lineages diverged. The earliest identifiable photosynthetic eukaryotic fossil was named *Bangiomorpha pubescens* based on its multicellularity, cell division pattern, and morphological resemblance to modern members of the red algal Bangiales [57]. Its age is currently estimated as 1.047 Gya (billion years) based on Re-Os isotope ratios [58]. Fossils of *Bangiomorpha* have been found in several localities in the Canadian Arctic Archipelago, where it lived in shallow seas; presumably in a high light environment but with O_2_ levels much lower than today. It is safe to assume that this alga had already developed mechanisms for dealing with the conflicting demands of photosynthesis and photoprotection.

Unfortunately, identifiable green algal fossils do not appear in the rock record until about 800 Mya. There are many well-dated Neoproterozoic single-celled fossils, classed as eukaryotic based on size and ornamentation, but there is simply no way to tell if they were photosynthetic or to what lineage they belonged (reviewed in [57]). Estimates of when the red-green split occurred are based on molecular clock analysis, which involves phylogenetic trees calibrated with a few dated fossils (some of them controversial) and molecular biomarkers (some equally debatable) [57]. Whatever the timing of the split, the PBS could have been lost much later, in a single branch of the green lineage which happened to be the only one to survive the glaciations and mass extinctions of the following billion years.

What we can say is that red and green algae independently evolved multicellularity and a variety of adaptive strategies that have allowed a few of their descendants to continue to thrive on the modern earth. The ancestral chlorophytes that lost the PBS were able to survive its loss because they already had members of the Chl-carotenoid-binding LHC family to take its place. Subsequent expansion of that family and evolution of a variety of additional stress-response mechanisms, but probably not the loss of the PBS, enabled some of their descendants to adapt to the extremely demanding land environment.

## 6. PBS Loss after Secondary Endosymbiosis

All the algal groups that obtained their plastids by secondary endosymbiosis of a red alga (heterokonts, haptophytes, cryptophytes, and apicomplexans) also lost their PBSs. Only one trace remains, the PE -β type subunit that forms part of the novel phycobiliprotein antenna found only in cryptophyte algae [59]. This suggests that at least in the Cryptophyta, the red algal chloroplast still had a PBS for some time after the endosymbiotic relationship was first established. In this special issue, Obornik has reviewed the debate over whether all four algal groups are the result of one secondary endosymbiosis, or several independent ones, or a combination of secondary and tertiary endosymbioses [32]. The evolutionary picture is further complicated by the fact that all four of these groups have acquired significant numbers of their nuclear genes from other algae and various bacteria, as well as the red endosymbiont(s).

An obvious explanation for the secondary loss(es) of the PBS is that genes for the nuclear-encoded linker proteins and enzymes needed for bilin synthesis and attachment could not be easily transferred to the host nucleus, and/or it was too difficult to retarget these proteins back across the four membranes now surrounding the plastid. However, there are many red algal nuclear genes for other plastid proteins that have been successfully transferred. There is also good evidence for several ongoing transfers of red plastid genes to the host nucleus, with successful retargeting of the encoded protein back to the plastid after endosymbiosis [60,61]. It seems that establishing a secondary or tertiary endosymbiosis is much easier than establishing a primary one, and that endosymbiogenesis by itself is unlikely to have been a driver of secondary PBS loss.

All of these groups, except for the cryptophytes, have very large families of Chl *a/c* Lhcs that diversified from the ancestral Lhcr’s independently of the Chl *a/b* branch of the family [40,41]. They have very active xanthophyll cycles, using the reversible interconversion of diadinoxanthin and diatoxanthin rather than violaxanthin and zeaxanthin. In response to high light and other stresses they induce Lhcx proteins, members of the LHC family related to the green algal LHCSRs, at least some of which are involved in aspects of photoprotection (reviewed in [50,53]). These factors would certainly have facilitated the eventual loss of the PBS.

The cryptophytes are much more like the red algal endosymbiont, in that they have a small family of Lhcr-like Chl *a/c* proteins and do not have a xanthophyll cycle [59,62]. They have evolved a unique light-harvesting antenna: a tetramer made up of two of the red algal β subunits plus two small “α” subunits that are totally unrelated to the PBS α subunits [63]. This new antenna is located in the thylakoid lumen and transfers energy to both PSI and PSII [64], but does not appear to be involved in energy dissipation, which occurs in the Lhcr antenna [65,66]. Did the acquisition of the novel α subunit gene (from a so-far unknown source) contribute to the dismantling and eventual loss of the PBS, or did this new protein just happen to ensure the retention of the β subunits by binding to them and forming a useful complex? We still know very little about the physiology of the cryptophyte algae; future research should cast more light on this question.

## 7. Conclusions

The PBS has been lost several times in evolution. In the case of the Chlorophyta, the loss appears to have been a question of chance, contingent on the prior development of an alternative light-harvesting antenna system binding chlorophylls rather than phycobilins. Subsequent to the loss of the PBSs, these antenna systems expanded and diversified, and evolved a variety of ways to offset the damaging effects of excess light energy. In the case of the prochlorophytes and *Acaryochloris*, chance and the Chl-proteins of the IsiA family might also have allowed the PBS to be lost. Losses subsequent to secondary endosymbiosis would certainly have involved expanded LHC families, but in the case of the cryptophytes, a fragment of the PBS was left behind and incorporated into a novel light-harvesting antenna.

## Figures and Tables

**Figure 1 biomolecules-09-00748-f001:**
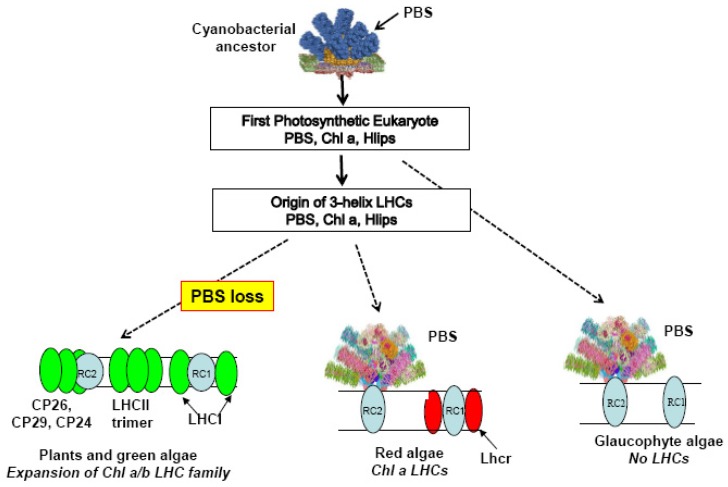
Evolution of light-harvesting antennas. RC1, Photosystem I core complex; RC2, Photosystem II core complex. Colored ovals, members of LHC superfamily. PBS, phycobilisome.

**Figure 2 biomolecules-09-00748-f002:**
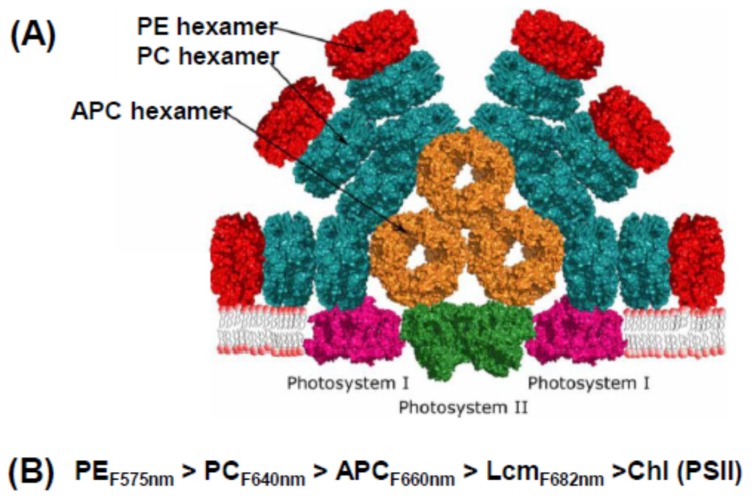
Phycobilisome and energy flow. (**A**) Model of a typical cyanobacterial phycobilisome as part of a megacomplex that includes PSI as well as PSII. Adapted from [17]. (**B**) Downhill energy cascade from PE to PSII reaction center. Numbers represent typical fluorescence emission maxima.

**Figure 3 biomolecules-09-00748-f003:**
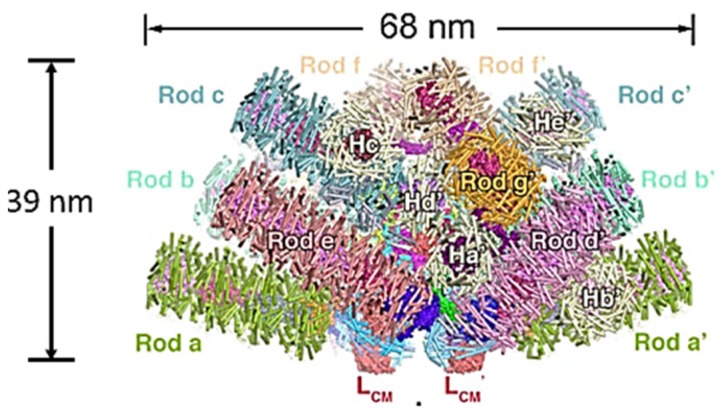
Structure of *Griffithsia pacifica* phycobilisome at 3.5Å resolution. View parallel to membrane plane. Rods are individually colored. Linkers cannot be seen from the surface. Adapted from Zhang et al. [21].

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
