# Peer review of "What Happened to the Phycobilisome?"

_biomolecules, 2019, doi:10.3390/biom9110748_

Round 1
Reviewer 1 Report
The review explored some of the reasons why the phycobilisome disappeared in plants and green algae while other species (red algae and glaucophyte algae) kept this structure. It also provided a little review of its structure and function regarding light absorption and photosynthesis.
Figure 1 legend is very small therefore it is hard to read.
In lines 97, 142, and 283 the author wrote the sentences using first person. The article should be written in the third person.
A minor English review is required (e.g, line 206).
Author Response
I thank the reviewer for catching some mistakes, which have been corrected.
However, I totally disagree with the idea that I should not use the first person. It was my idea, my hypothesis, and my discussion of the reasons I proposed that hypothesis.
Reviewer 2 Report
This is a very intriguing review article starting with a simple but fundamental question "What happened to the phycobilisome (during the course of evolution)?" The author discussed this problem based on the composition of the antennae system and photoprotection mechanism of Cyanobacteria, Chlorophyta, Rhodophyta, Glaucophyta and Cryptophyta, and concluded that the prior development of an alternative light-harvesting antenna system binding chlorophylls might have allowed the phycobilisome to be lost.
For more extensive discussion, I suggest addition of some information listed below.
Please describe more about the extent of conservation of constitutive/Fe-inducible Pcb-IsiA chlorophyll antenna among cyanobacterial species. Do most cyanobacterial species possess Fe-inducible Pcb-IsiA antenna? Is there any species having both PBS and constitutive Pcb-IsiA antenna, or is constitutive expression of Pcb-IsiA antenna always linked with loss of PBS?Please discuss appropriateness (or advantage/disadvantage) of antennae systems employed by each photosynthetic organism from the view of the range of wavelengths they can absorb and light environment of their growth habitat.
Information on carotenoid species and major photoprotection mechanism employed by each photosynthetic organism is better to be presented as figure or table. I think this information can be added to Fig.1.
Reviewer 3 Report
Phycobilosomes are very intricate light harvesting complexes that work like a funnel for photons. This is in stark contrast to all other known antenna complexes where the exciton is migrating along a random path. The question Beverley Green raises in her review why the PBS got lost during evolution in some photosynthetic lineage is therefore exciting.
As I understand she mainly argues that evolutionary contingency was a major cause for the loss of PBS. To undergo the loss the respective lineages had to develop a small LHC family before, which was already at hand in their cyanobacterial ancestor because of the HLIPs. The latter just had to undergo a duplication to form a protein very similar to today ´s LHCs.
From this argumentation it seems that the PBS got lost just because evolution follows a random path.
Although the idea might be correct there are also a lot of arguments against PBS in organisms like green algae and land plants. I think it would be nice to discuss at least a few. From my point of view there are e.g.
Organisms living in high light environments (like green algae and land plants) might get along better without PBS just because they are such good light harvesting devices. Too much excitation energy could be a selective disadvantage since relatively more protective measures against ROS are necessary and consumed. Without PBS grana thylakoids can form. This is a unique feature of all chlorophytes. The reason of the formation of the stacks is still debated but there are different possible explanations. One is the segregation (separation) of fast (PSI) and slow (PSII) photosystems, avoiding a spill-over of excitation from on to the other and regulating the excitation energy distribution between the two systems (e.g. Trissl and Wilhelm 1993). Another is that it has a protective role for PSII under high light (Anderson and Aro 1994).Minor issues:
Fig. 3 it says “perpendicular to the membrane place” but I guess it should be “membrane plane” but it looks as if it is parallel to the membrane plane.
p. 8 line 226 what is “refs]”?
